# Enhancement of VGG16 model with multi-view and spatial dropout for classification of mosquito vectors

Wanchalerm Pora[1]*, Natthakorn Kasamsumran[1‡], Katanyu Tharawatcharasart[1‡], Rinnara Ampol[2‡], Padet Siriyasatien[2‡], Narissara Jariyapan[2]*

**1** Department of Electrical Engineering, Faculty of Engineering, Chulalongkorn University, Bangkok, Thailand, **2** Department of Parasitology, Center of Excellence in Vector Biology and Vector-Borne Disease, Faculty of Medicine, Chulalongkorn University, Bangkok, Thailand

☯ These authors contributed equally to this work.
‡ NK, KT, RA and PS also contributed equally to this work.
* wanchalerm.p@chula.ac.th (WP); narissara.j@chula.ac.th (NJ)

**Data Availability Statement:** All Python codes and simulation outputs are accessible on GitHub (https://github.com/Wanchalerm-Pora/

## Abstract

Mosquitoes transmit pathogens that can cause numerous significant infectious diseases in humans and animals such as malaria, dengue fever, chikungunya fever, and encephalitis. Although the VGG16 model is not one of the most advanced CNN networks, it is reported that a fine-tuned VGG16 model achieves accuracy over 90% when applied to the classification of mosquitoes. The present study sets out to improve the accuracy and robustness of the VGG16 network by incorporating spatial dropout layers to regularize the network and by modifying its structure to incorporate multi-view inputs. Herein, four models are implemented: (A) early-combined, (B) middle-combined, (C) late-combined, and (D) ensemble model. Moreover, a structure for combining Models (A), (B), (C), and (D), known as *the classifier*, is developed. Two image datasets, including a reference dataset of mosquitoes in South Korea and a newly generated dataset of mosquitoes in Thailand, are used to evaluate our models. Regards the reference dataset, the average accuracy of ten runs improved from 83.26% to 99.77%, while the standard deviation decreased from 2.60% to 0.12%. When tested on the new dataset, *the classifier*'s accuracy was also over 99% with a standard deviation of less than 2%. This indicates that the algorithm achieves high accuracy with low variation and is independent of a particular dataset. To evaluate the robustness of *the classifier*, it was applied to a small dataset consisting of mosquito images captured under various conditions. Its accuracy dropped to 86.14%, but after retraining with the small dataset, it regained its previous level of precision. This demonstrates that *the classifier* is resilient to variation in the dataset and can be retrained to adapt to the variation. The *classifier* and the new mosquito dataset could be utilized to develop an application for efficient and rapid entomological surveillance for the prevention and control of mosquito-borne diseases.

Mosquitoes-TH/tree/main/iPyNB). The dataset produced in this study can be publicly accessible via the URL: https://www.kaggle.com/datasets/cyberthorn/chula-mosquito-classification.

**Funding:** This study is funded by Thailand Science, Research and Innovation Fund Chulalongkorn University (CU_FRB65_hea(45)_052_30_33). This fund is managed by Thailand science, research and innovation office (https://www.tsri.or.th). Its principal investigator and investigator are Narissara Jariyapan and Wanchalerm Pora, respectively. The funders had no role in study design, data collection and analysis, decision to publish, or preparation of the manuscript.

**Competing interests:** The authors have declared that no competing interests exist.

# Introduction

Several species of mosquitoes are responsible for the transmission of various pathogens to humans and animals [1]. For example, *Aedes aegypti* (the yellow fever mosquito) is a vector of the yellow fever virus, dengue virus, chikungunya virus, and zika virus, whereas *Aedes albopictus* (the Asian tiger mosquito) transmits chikungunya virus, dengue virus, and dirofilaria parasites and is considered a potential vector of the zika virus [2]. *Aedes vexans* is an important vector that transmits West Nile virus (WNV) from birds to mammals [3]. *Culex quinquefasciatus* (the southern house mosquito) is a vector of the filarial nematode, *Wuchereria bancrofti*, Rift Valley fever virus, and WNV [4]. *Culex vishnui* is the most common vector of Japanese encephalitis virus (JEV) in South-East Asia [5]. *Anopheles tessellatus* has been found naturally infected with human malaria parasites in Indonesia [6], Sri Lanka [7], along with JEV in Taiwan [8]. Both entomological surveillance and the monitoring of mosquitoes are very important for the prevention and control of mosquito-borne diseases. The timely detection of vector species can save many lives and reduce economic losses.

The classification accuracy of traditional models, as represented by the support vector machine (SVM) [9], and those of a few deep learning models, including AlexNet, VGGNet, and ResNet, have been carried out [10]. As for SVM, twelve types of features had to be extracted beforehand. The three deep neural networks, however, did not need feature extraction. The maximum accuracy of SVM was 82.4%, and that of the deep learning models was found to be up to 13.1% higher.

The effectiveness of the three convolutional neural networks (CNN): AlexNet [11], LeNet [12], and GoogLeNet [13] was evaluated [14]. Results demonstrated that GoogLeNet yielded the best result, having an accuracy of 76.2%. AlexNet and LeNet attained an accuracy of 51.2% and 52.4%, respectively. This work suggests that networks with more complexity are likely to be more accurate.

Regarding the mosquito classification, conventional CNN models such as VGG16, ResNet, and SqueezeNet with data augmentation and transfer learning have achieved state-of-the-art results. The oldest VGG16 yielded the best results. Furthermore, this work revealed that morphological keys used by human experts have been found to be similar to the heatmap of the CNN viz. patterns of the body, legs, and proboscis. This work published an image set, including five classes of disease vectors and one class of non-vectors [15]. Thereafter, we would refer to it as *the reference dataset*.

Further, two deep convolutional neural networks (DCNN), YOLOv3 [16] and Tiny YOLOv3, were deployed to classify five vector classes and a non-vector class. This work [17] reused images from *the reference dataset* along with additional unpublished non-vector images. It is seen that YOLOv3 and Tiny YOLOv3 achieved 97.7% and 96.9% accuracy, respectively. Subsequently, the authors amplified the performance of the YOLOv3 model using four augmentation conditions. Results showed that when *default* + blur + noise augmentation were used, such conditions increased the accuracy of classification up to 99.1%.

Recently, two DCNN models including VGG16 [18] and ResNet50 [19] with initial weights from ImageNet [20], were investigated in [21]. The models reused the dataset published [22]. Two of the networks i.e. VGG16 and ResNet50 were employed to separate *Aedes albopictus* species from non-*Aedes albopictus*. Thus, it was found that VGG16 achieved more than 94% accuracy on the test set. Furthermore, the authors identified that the white band stripes in the legs, the abdominal patches, the head, and the thorax were features used by *the classifier*. A heatmap visualization of discriminative regions was also provided. This recent work [21] found that VGG16 again yielded the best result [15].

It is evident that the VGG16 model appears to be the most promising, as it provided the best results. Moreover, in situations where the quality of images cannot be guaranteed, model robustness is also crucial. In the method section, *the classifier* is composed of three multi-view models (A), (B), and (C), as well as an ensemble model (D) to predict mosquito species from three images rather than one. Note that photographing a mosquito three times does not require much more work than photographing it once. We assume that by applying three images, *the classifier* can produce more accurate and more robust results.

## Materials and methods

### Ethics statements

This study was approved by the animal research ethics committee of Chulalongkorn University Animal Care and Use Protocol (CU-ACUP), Faculty of Medicine, Chulalongkorn University, Bangkok, Thailand (COA No. 029/2564).

### Sources of mosquitoes and species identification

*Ae. aegypti, Ae. albopictus*, and *Cx. quinquefasciatus* mosquitoes have been maintained continuously for many consecutive generations in an insectary at the Department of Parasitology, Faculty of Medicine, Chulalongkorn University, Bangkok, Thailand. Some of them were used in this study. Live engorged female mosquitoes were collected from six provinces of Thailand, including Chiang Mai (18.891369, 99.112922; Northern), Bangkok (13.7253188, 1007529578; Central), Chai Nat (15.2233373, 100.1907276; Central), Pratum Thani (14.0619100, 1005437231; Central), Samut Prakan (13.5770855, 1008557932; Central), and Nakhon Si Thammarat (8.2169321, 99.8872206; Southern), from October 2021 to May 2022. Each of them oviposited eggs in an isolated ovipot to establish an iso-female line in the insectary using the methods described [23, 24]. To identify species, a representative of the F1 progeny of each isoline was subjected to morphological and molecular identification using keys published [1, 25, 26] and a PCR method using the barcoding primers of the mitochondrial cytochrome c oxidase subunit I (*COI*) gene, LCO1490 (5′–GGT CAA CAA ATC ATA AAG ATA TTG G–3′) and HCO2198 (5′–TAA ACT TCA GGG TGA CCA AAA AAT CA–3′) [27]. PCR products were purified using the PureLink™ PCR Purification Kit (Thermo Fisher Scientific, Vilnius, Lithuania) and sequenced using a 23 ABI 3730XLs sequencer (Macrogen Inc., Seoul, South Korea). To generate the image dataset, 231 *Ae. aegypti*, 223 *Ae. albopictus*, 197 *Ae. vexans*, 160 *Cx. quinquefasciatus*, 269 *Cx. vishnui*, 178 *An. tessellatus*, and 236 *Cx. aculaetus* females, or about 200 specimens each from the F1 progeny, were sampled. Additionally, 100 females of each laboratory strain, including *Ae. aegypti, Ae. albopictus*, and *Cx. quinquefasciatus*, were used. Each and all mosquitoes were photographed a few times. All photos would then be pre-processed, and the resulting images would comprise an image collection that would henceforth be referred to as *the new dataset*.

### Image datasets

Two image datasets were utilized: (1) *the reference dataset* of mosquitoes in South Korea [15] and (2) *the new dataset* of mosquitoes in Thailand (generated in this study). Using the sources described previously, the dataset containing images of mosquitoes in Thailand was generated in two distinct settings.

In the first setting, 4,590 high-quality photographs were taken using a flagship Vivo V21 (Guangzhou, China) smartphone equipped with a phone holder, an LED ring light, and a clip-on macro lens, as depicted in Fig 1. The 64MP smartphone was set to produce photos at its

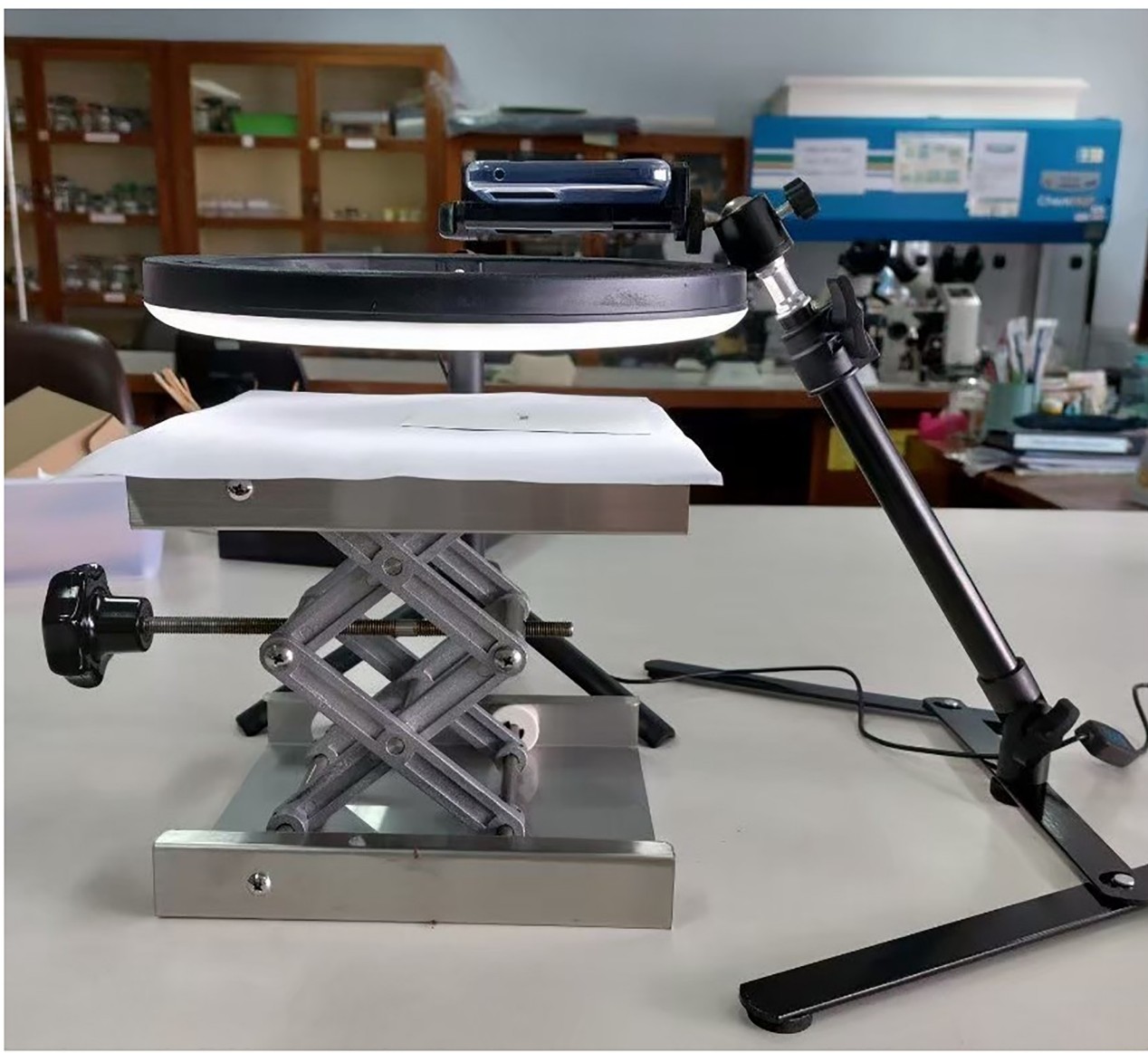

**Fig 1. One of the two settings for photography.** A proper holder was used to attach a high-end (VIVO-V21) camera phone equipped with a clip-on macro lens and an LED ring light. A mosquito was placed approximately 7 cm (the shortest focusable distance) below the phone's camera. The smartphone was triggered by a remote control in order to minimize vibration and ensure high-quality photos. In another setting, mosquitoes were photographed from a variety of angles using a midrange (Samsung Galaxy A52s) or subpar (Vivo Y21) camera phone without the aid of holders or lighting. Due to blurring caused by hand movement, shading caused by flash lighting, or higher noise caused by a low-quality sensor, photos taken in such a way turn out to be of low quality.

highest resolution of 9,248 × 6,936. The clip-on lens was used to shorten the focus distance, magnifying the mosquito in the photograph. This collection of images was used to evaluate the accuracy of our proposed model.

In the second setting, ten or more mosquitoes per species were resampled. They were captured by mid-range smartphones: Samsung A52s (Seoul, South Korea) and a low-end Vivo Y21, which produced 459 and 564 images, respectively. The photographer held one of the mobile devices with the same clip-on lens while photographing mosquitoes under variable indoor lighting conditions from different angles. The 64MP mid-range and the 13MP low-end

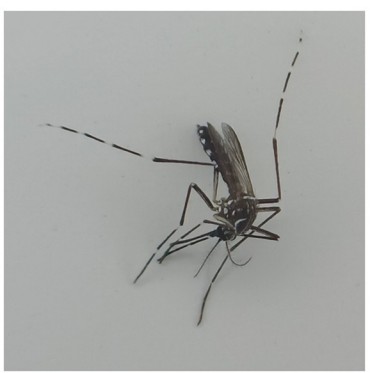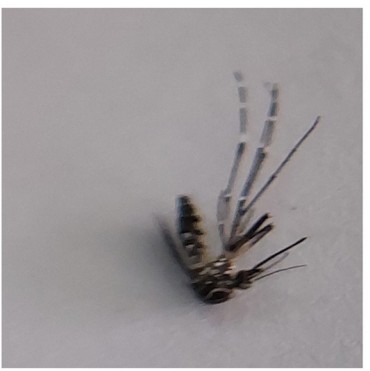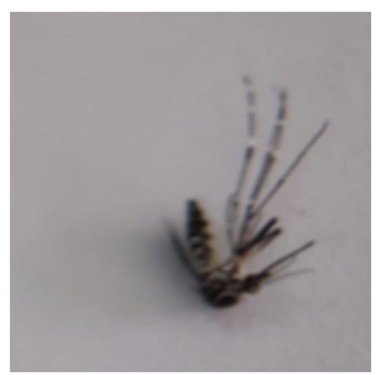

**Fig 2. Cropped and resized mosquito images.** Having the correct setting and a high-quality camera, the 512x512 images are of excellent quality (LHS). Without a holder, adequate lighting, or a camera of sufficient quality, the resulting image may be blurry with high shading and noise levels (middle). Blurry photographs reduce the accuracy of all classification models (RHS). The picture on the right was purposely blurred from the one in the middle.

phones produced images with resolutions of 9,248 × 6,936 and 4,128 × 3,096, respectively. Note that the quality of the lens and image sensors on the phones' cameras has a greater impact on the resulting photos than the resolutions of the cameras. In this simple setting, the photographs generated may be blurry, feature shading, or have a high level of shot noise. This second collection of photos was utilized to assess the robustness of the proposed models and to evaluate the results of retraining. The difference in the quality of images between the first and second settings can be clearly seen (left and middle images) in Fig 2. Table 1 lists the number of females and photos captured by three smartphones from each biological species, along with their collection sites.

Despite our efforts to minimize the distance between the camera and the subject, the mosquitoes in the photographs are quite small. A YOLOv5 [28] that had been trained to estimate the boundaries of mosquitoes (if any) was deployed to crop and resize each image to 512 × 512 pixels. All of the photos captured in the first setting were successfully cropped and resized. Of about one thousand photos taken in the latter setting, only 31 photos (3%) were found to be of no use due to their very low quality (the YOLOv5 could not find a mosquito in it), and thus were discarded.

## Techniques to improve accuracy

**Transfer learning.**  Transfer learning is a machine learning technique in which the parameters of a well-trained model are used as the initial parameters of another model before it is

**Table 1. Biological mosquito species, collection sites, and the number of females/images taken in various lighting conditions by three different smartphones.**

| *Species* (GenBank accession no.) | Collection sites | No. of females/images captured by | | |
|---|---|---|---|---|
| | | **Vivo V21** | **Samsung A52s** | **Vivo Y21** |
| *Ae. aegypti* (OP477050–OP477052) | Laboratory, Chiang Mai, Bangkok, Samut Prakan, Nakhon Si Thammarat | 321/763 | 10/61 | 10/101 |
| *Ae. albopictus* (OP477053–OP477055) | Laboratory, Chiang Mai, Bangkok, Chai Nat, Nakhon Si Thammarat | 323/666 | 10/62 | 10/82 |
| *Ae. vexans* (OP477062–OP477064) | Chiang Mai | 197/591 | 15/89 | 11/66 |
| *An. tessellatus* (OP477047–OP477049) | Chiang Mai | 178/533 | 10/59 | 10/72 |
| *Cx. quinquefasciatus* (OP477056–OP477058) | Laboratory, Chiang Mai, Bangkok, Samut Prakan, Nakhon Si Thammarat | 260/719 | 11/68 | 10/72 |
| *Cx. vishnui* (OP477059–OP477061) | Chiang Mai | 269/610 | 10/60 | 10/85 |
| **Other Mosquitoes** | Pathum Thani | 236/708 | 10/60 | 10/86 |
| **Total** | | **1,784/4,590** | **76/459** | **71/564** |

trained, provided that both models have identical or comparable structures. *ImageNet* offers multiple sets of parameters for well-known models, including the VGG16. In this study, ImageNet's parameters were used to begin training all of our modified VGG16-based models when high-quality images from *the reference dataset* and from *the new dataset* were applied. Later, when the low-quality images from *the new dataset* were applied, the parameters obtained from our trainings (using the high-quality images) would be transferred to retrain the models.

**Data augmentation.** It is well-known that, as the amount of data increases, the performance of the deep learning model also improves. In general, data augmentation is a technique of altering existing data to create a lot more data for model training purposes. As the number of mosquito specimens in our lab are limited, only 600 *original* images generated from photos taken in our well-lit lab were available. We employed data augmentation or *image augmentation* [29] to expand the original dataset *artificially* to train our deep neural networks. Herein, the augmented images are representative of the quality of images that we get from users who take photos in various conditions.

- **Random zoom**. Due to the fact that YOLOv5 was already deployed to resize images, so that they all have the same dimensions of 512 by 512 pixels, the mosquitoes on the images are not proportionally scaled. In addition, lab-reared mosquitoes are typically larger than those found in their natural habitat due to superior nutrition. Therefore, size cannot ever be a defining characteristic. We want each model to observe mosquitoes of varying sizes. All images of mosquitoes were randomly resized to within ±20% of their original dimensions. This method ensures that models are insensitive to the size of mosquitoes depicted in images. In other words, the models can learn the appearance of a species regardless of the size of the mosquitoes.

- **Image rotation**. Similar to variation in size, we cannot ask our prospective users to take photos of mosquitoes at a specific angle. Therefore, the image should be randomly rotated within a range of ±180° so that the models are invariant to the rotation of the mosquito images.

- **Random crop**. It is noted that YOLOv5 helps us to zoom in and resize the images of the mosquitoes. Most of the time, the bounding box is slightly off center. So after resizing, the mosquito appears near the center of the image (but not exactly there). In addition, $512 \times 512$-pixel images are kept in a database for future use; in this work, the images were resized further down to $256 \times 256$ pixels before use. Then, the images are randomly cropped, providing $224 \times 224$ pixels to let the models learn the mosquitoes while they move around the center of the images.

- **Random brightness**. Modern mobile phones usually handle the brightness of photos very well, especially in not-so-bad lighting conditions. Therefore, we randomly change the brightness of images only in a range of ±10%.

- **Random hue**. Users are advised to take photos in good lighting conditions. In terms of brightness, most users know what is "good". However, it is not the case for hue or color temperature; the background of images may appear reddish, yellowish, or blueish. Thus, it is sensible to randomly rotate the hue: fully 360° to let the models see the same mosquito images in various background colors.

- **Gaussian noise injection**. When taking photos, shot noise or image noise is always present, especially on mobile phones with low-quality cameras. In addition, it is well-known that training a deep model with noise can reduce the likelihood of overfitting, particularly when

the dataset is relatively small in comparison to the number of parameters. Before employing any image, we intentionally added random Gaussian noise with a standard deviation of 0.1.

**Spatial dropout.** Training deep learning models on relatively small datasets can lead to overfitting (to the training data, but not to the general data); training them for too many epochs guarantees that they will become overfit as well. In Fig 3, the architecture of VGG16 consisting of five convolutional blocks, is depicted. Each block contains two or three consecutive convolutional layers followed by a max pooling layer. The total number of parameters exceed 138 million. Even though the number of images have been increased from 4,600 to 20,000 via image augmentation described previously, the dataset is still relatively small. So there is still an opportunity for the model to overfit. To reduce the chance, a dropout layer may be inserted between two layers. In the training phase, it randomly omits values from nodes in the previous layer to nodes in the next layer, thereby ensuring that the output(s) of the models will never be an exact match for the dataset. Despite higher loss during training, greater accuracy is obtained during testing. It is noted that no values are dropped during testing or inference phase; rather, they are averaged out.

*Spatial dropout* [30] evolves from *dropout*. However, for images, dropout may not be effective enough due to the high mutual correlations between the adjacent 2-D pixels. In the case of classification, dropping any of them may produce the same outcome. Instead of blocking random nodes within and between channels, spatial dropout blocks all nodes of some random channels because correlation between channels is significantly lower than correlation within channels. The distinction between standard dropout and spatial dropout is shown, as in Fig 4.

In this paper, we propose to insert five spatial dropout layers, each of which is after the max pooling layer of each convolution block. The architecture of our modified VGG16, designated as "SDVGG16", is depicted in Fig 5.

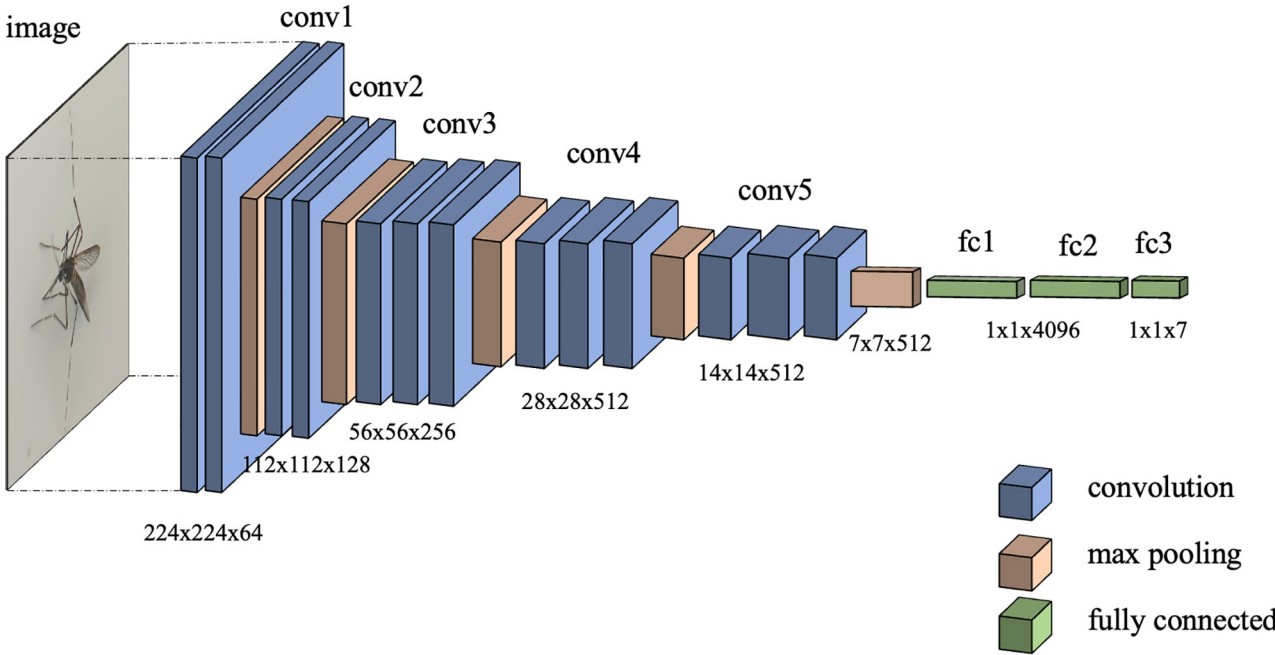

**Fig 3. Architecture of VGG16.** It is composed of five convolution blocks, each followed by a max pooling layer, and a block of three fully-connected layers.

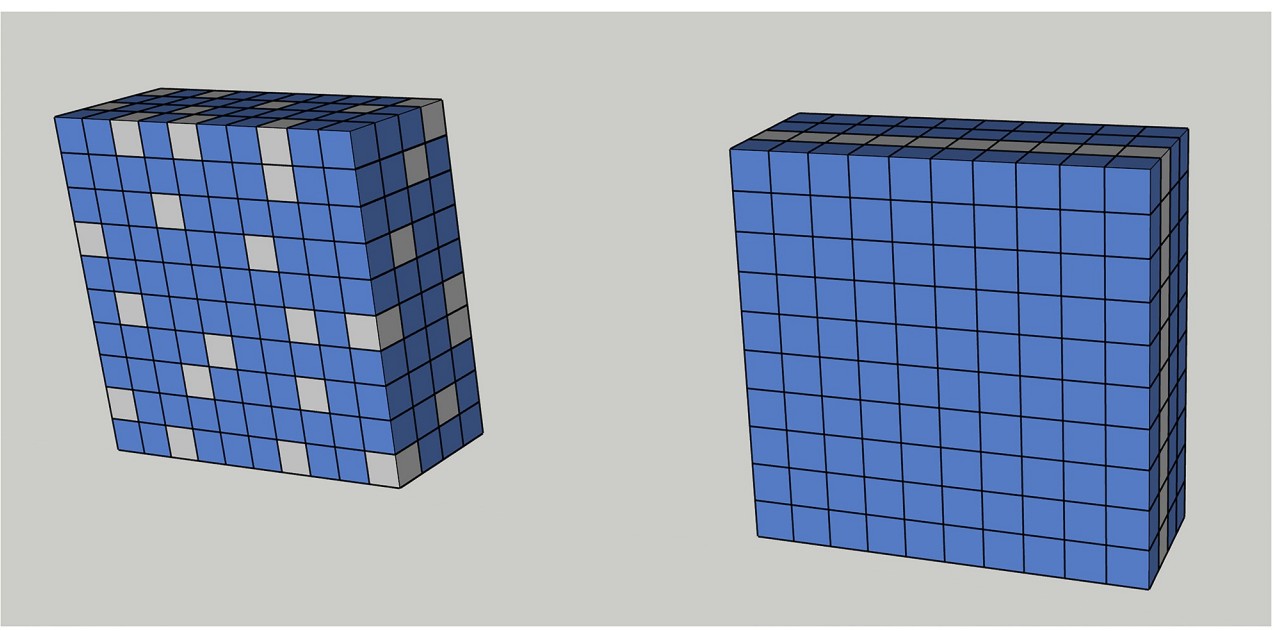

**Fig 4. Dropout vs. spatial dropout.** The conventional dropout (left) blocks some nodes (gray squares) in 3-D randomly; the spatial dropout (right) selects at random some channels first and blocks all the nodes in the chosen ones.

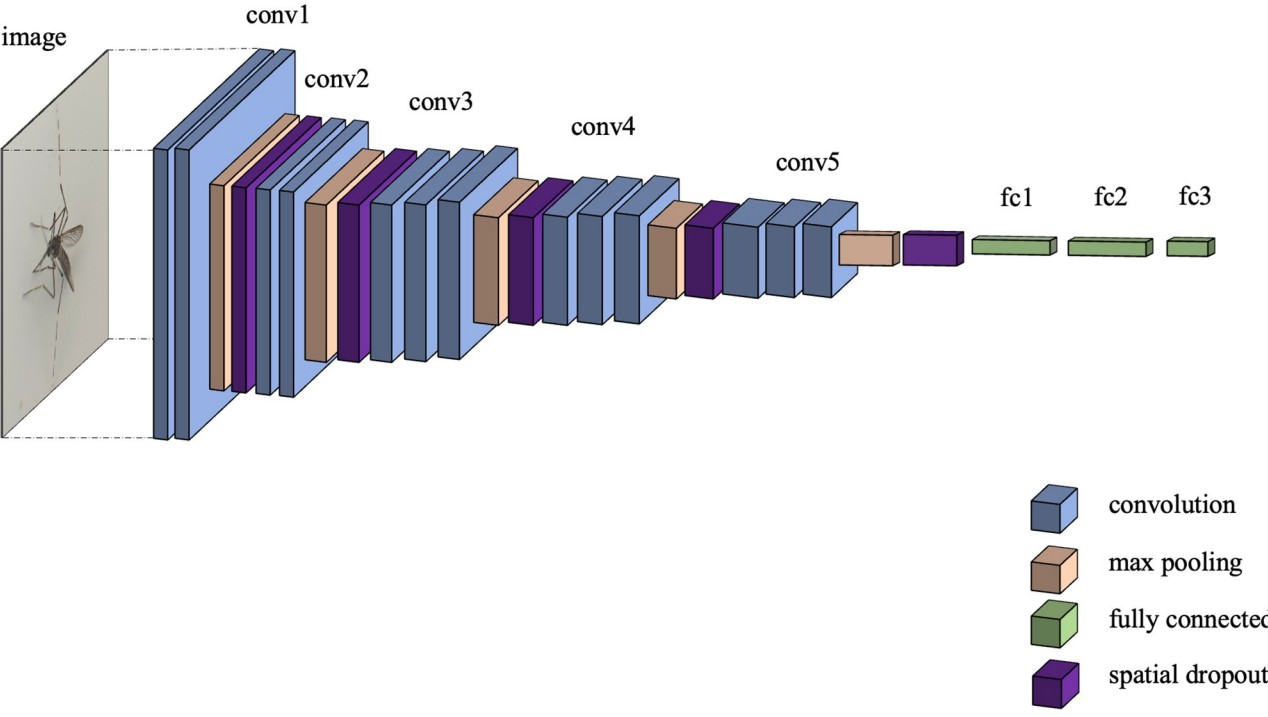

**Fig 5. Architecture of SDVGG16.** Modified from the VGG16, a spatial dropout layer is inserted after each of the five max pooling layers.

**Multiple inputs.** To increase the network's robustness, we propose employing models that receive two images rather than one, as is the case with the SDVGG16 network. If one image lacks the part(s) that specifies the species, or if one image is of poor quality, the models having two inputs would have a greater chance of identifying the correct species.

As shown in Fig 6, the dropout concept of the spatial dropout is expanded upon so that two layers from two input images can be merged into a typical VGG16 layer. It is seen that the odd channels of the first image are blocked, while the even channels of the second image are dropped. The unblocked channels are interleaved to form a layer with the same shape as the conventional single input model. This study focuses on the following three dual-view models:

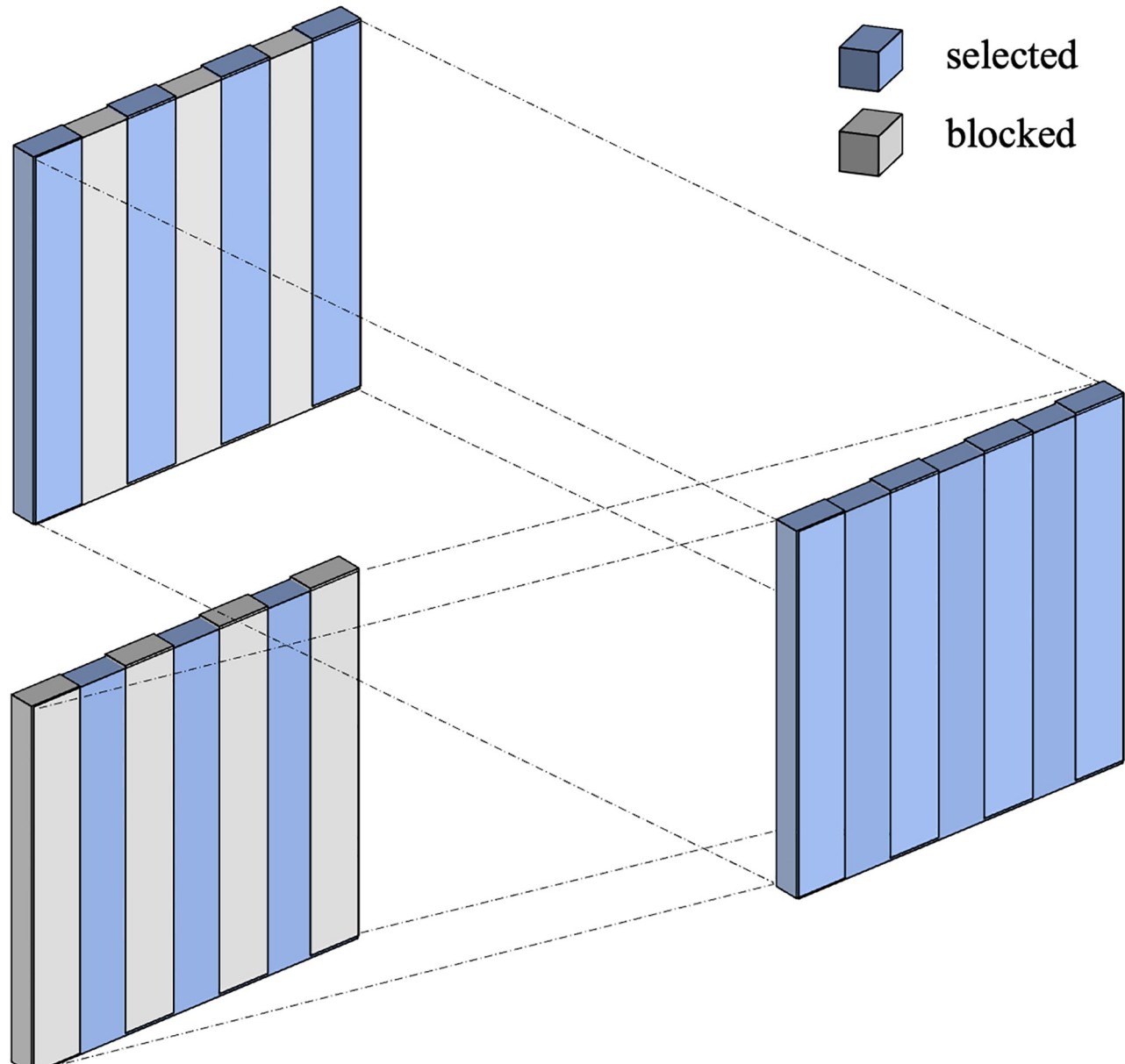

**Fig 6. Combining two inputs.** Adapted from the spatial dropout, channels from two images can be combined by interleaving the odd channels from one image, and the even one from the other.

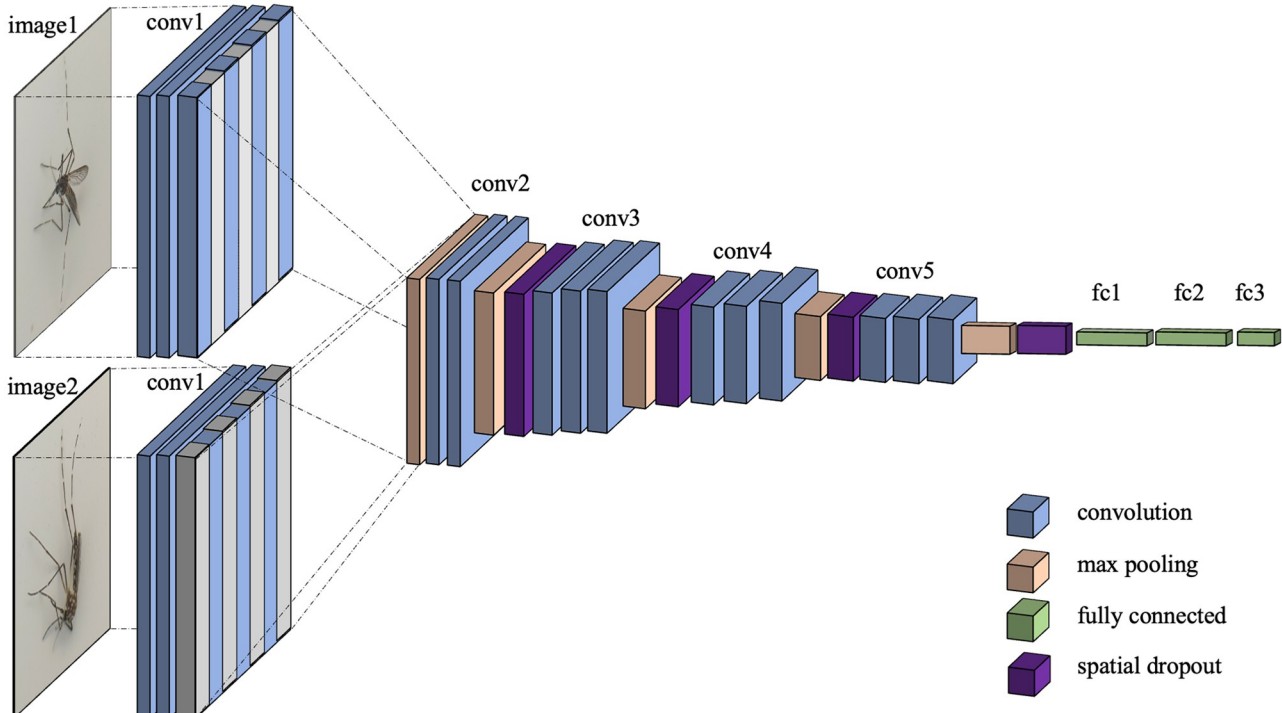

**Fig 7. Architecture of early-combined model (Model (A)).** Two branches from two image inputs are combined just after the first convolution block.

- **Early-combined model**. Two branches from two inputs are combined just after the *first* convolution block. Later, it may be referred to as Model (A).

- **Middle-combined model**. Two branches from two inputs are combined after the *third* convolution block. Later, it can be called Model (B).

- **Late-combined model**. Two branches from two inputs are combined after the *fifth* convolution block. It may henceforth be referred to as Model (C).

The architectures of the early-, the middle-, and the late-combined models are shown in Figs 7–9, respectively.

## Classifier

One of the principal outcome of this study is our *classifier* as illustrated in Fig 10. It is composed of the three dual-view models: Models (A), (B), and, (C) and an ensemble model described below. Our classifier emphasizes its robustness so it requires three images instead of one. All three images are sent to each and all three models. As such, this produces three pairs from three inputs, $\binom{3}{2}$. Hence, each model produces three prediction results, which are three sets of confidence scores. Totally, the three models give out nine sets of confidence scores.

**Ensemble model.** After the three dual-view models have predicted the species from three images, the accuracy of *the classifier* can be enhanced by fusing nine sets of confidence scores into one using an ensemble model, which may be henceforth called Model (D). Each dual-view model contributes three sets of confidence scores to the ensemble model. The species that receives the highest fused score is the optimal answer of both Model (D) and *the classifier*. In

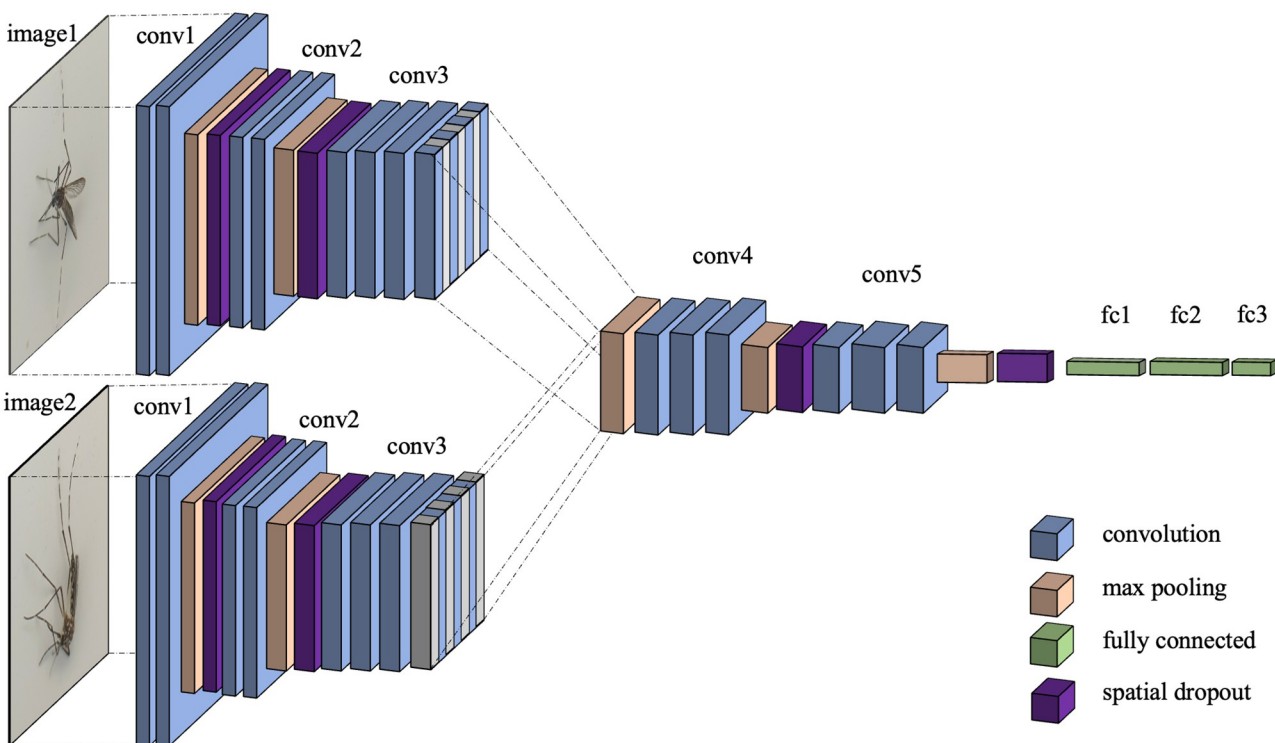

**Fig 8. Architecture of middle-combined model (Model (B)).** Two branches from two image inputs are combined after the third convolution block.

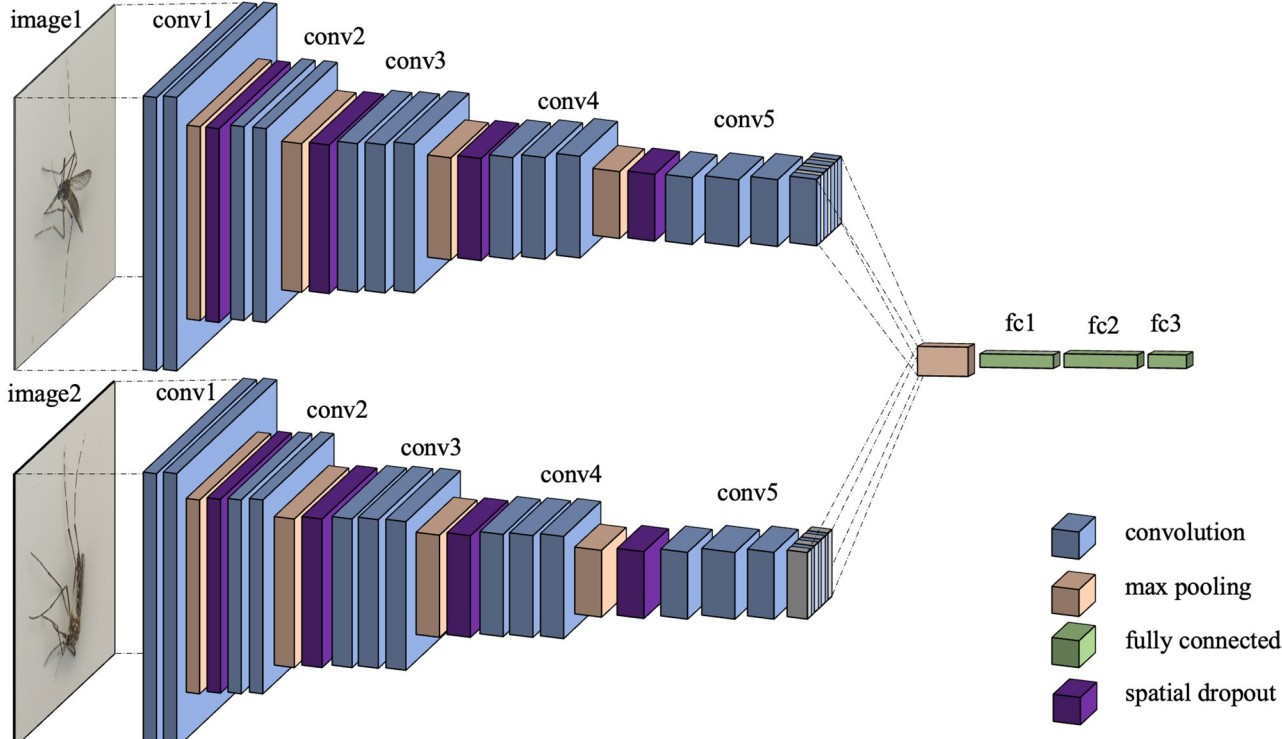

**Fig 9. Architecture of late-combined model (Model (C)).** Two branches from two image inputs are combined after the fifth convolution block.

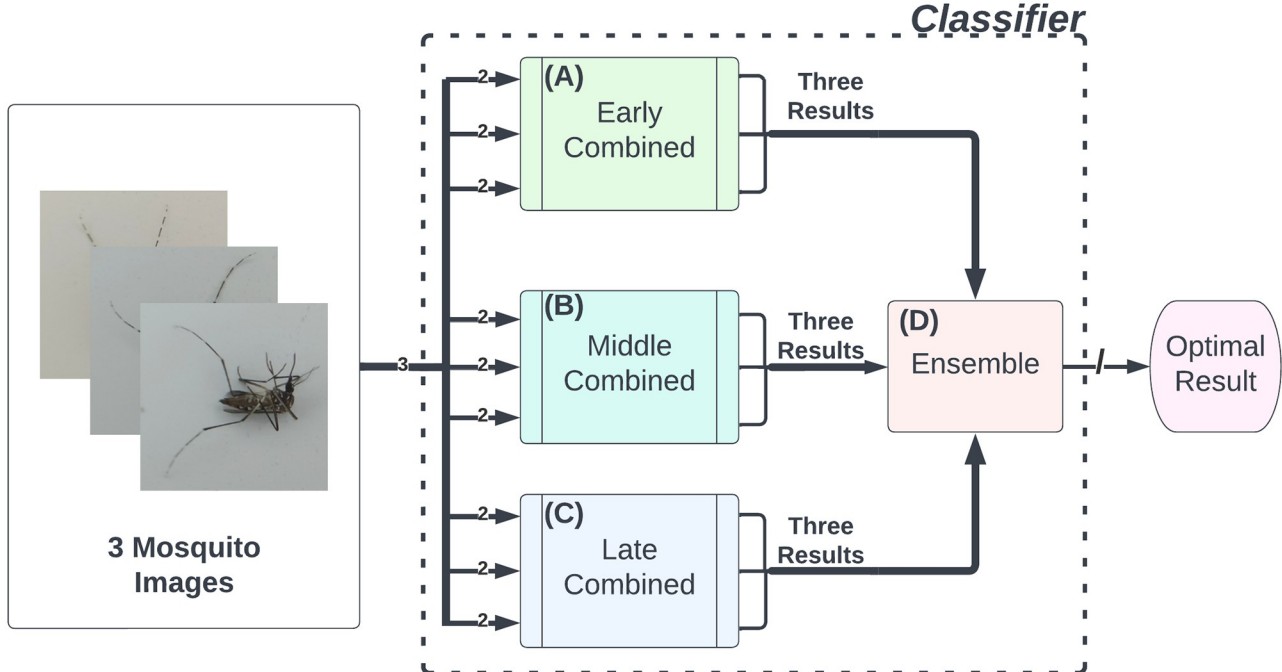

**Fig 10. *The classifier.*** It is composed of three deep learning models: Models (A), (B) and (C), which attempt to predict the species from three image pairs from three input images, and an ensemble model (Model (D)), which combines and yields the optimal classification result.

this study, we propose to make use of a simple but effective artificial neural network (ANN) with a 6-node hidden layer as the optimal result combiner.

## Results and discussion

Multiple experiments were designed to assess the performance of our proposed models. All the models were written in Python with the Keras API [31], which is built on top of the TensorFlow Framework [32]. KerasCV provides some image processing functions [33].

In all experiments, the same codes were executed ten times. Performance of the models was investigated in accordance with the scores gained via their means and standard deviations of their classification accuracy. Herein, results are provided in the following tables (Tables 2–8)). The best, second-best, and worst of means and standard deviations are indicated by the figures in bold, italic, and underline, respectively.

Our Python codes and simulation outputs in the form of the well-known Jupyter notebook are available on GitHub, whose link is given in S1 Notebooks. The link to download the dataset produced in this study can also be found on S1 Dataset.

### Experiments on the reference dataset

A couple of experiments utilizing *the reference dataset* have been set up. The first one attempts to determine the optimal spatial dropout rate. The second experiment aims to evaluate the precision and the robustness of the proposed models. In the same article that published *the reference dataset*, a Python code for its VGG16 version, henceforth referred to as *the reference model*, was also given [15]. All the Python codes for all models were carried out ten times. At the beginning of each run, the dataset was randomly divided into 8:2 proportions of training

**Table 2. Mean and standard deviation of classification accuracy of *the reference model* in comparison to those of the SDVGG16 model with the dropout rate from 0.1 to 0.6 on *the reference dataset*.**

| Run | Accuracy (%) when dropout rate = | | | | | | |
|---|---|---|---|---|---|---|---|
| | *reference* | 0.1 | 0.2 | 0.3 | 0.4 | 0.5 | 0.6 |
| 1 | 84.39 | **98.35** | 98.17 | *98.28* | 97.13 | 95.37 | 92.21 |
| 2 | 82.30 | **99.24** | *98.48* | 98.17 | 97.20 | 95.80 | 91.62 |
| 3 | 83.65 | *98.48* | 98.29 | **98.50** | 97.30 | 94.38 | 91.02 |
| 4 | 80.34 | **99.27** | *98.74* | 98.15 | 96.49 | 96.15 | 90.89 |
| 5 | 81.86 | **98.31** | 98.18 | *98.26* | 97.70 | 95.79 | 91.67 |
| 6 | 89.05 | *98.54* | **98.88** | 97.69 | 97.26 | 95.65 | 93.58 |
| 7 | 82.23 | **98.83** | *98.50* | 97.79 | 97.47 | 96.08 | 89.37 |
| 8 | 80.00 | **98.98** | *98.52* | 98.06 | 96.70 | 95.75 | 88.67 |
| 9 | 82.53 | *98.74* | 98.67 | **97.80** | 97.14 | 93.89 | 91.86 |
| 10 | 86.28 | *98.15* | **98.39** | 96.35 | 97.68 | 95.38 | 91.28 |
| Mean | 83.26 | **98.69** | *98.48* | 97.91 | 97.21 | 95.42 | 91.22 |
| SD | 2.602 | *0.369* | **0.223** | 0.571 | 0.3646 | 0.696 | 1.319 |

Figures in bold and italic indicate the best and second-best values in the row, whereas figures with an underline show the worst values.

and test datasets; the models were initialized with parameters from ImageNet. It is noted that when either an augmented or original image was selected for the training dataset, all of the augmented/original versions were chosen too, due to their high degree of correlation.

**SDVGG16.** In this experiment, the dropout rate of the SDVGG16 model varied from 0.1 to 0.6 with a spacing of 0.1. Results were then compared with those of *the reference model* that had no spatial dropout layer. Their classification accuracy gained from ten executions are shown in Table 2. As expressed in Table 2, the accuracy of *the reference* model are significantly lower than those stated in its article [15]. It is claimed that the *fine-tuned* accuracy is 97.19%,

**Table 3. Mean and standard deviation of classification accuracy of *the reference model* in comparison to those of our multi-view models on *the reference dataset*.**

| Run | Accuracy (%) | | | | |
|---|---|---|---|---|---|
| | *reference* | (A) | (B) | (C) | *Classifier* |
| 1 | 84.39 | 97.08 | 97.73 | *98.61* | **99.70** |
| 2 | 82.30 | 98.10 | 97.16 | *98.35* | **99.83** |
| 3 | 83.65 | 97.58 | 97.41 | *98.82* | **99.79** |
| 4 | 80.34 | 98.84 | 98.81 | *99.24* | **99.96** |
| 5 | 81.86 | 98.87 | 98.51 | 97.66 | **99.79** |
| 6 | 89.05 | 98.25 | 98.13 | *98.64* | **99.70** |
| 7 | 82.23 | 98.27 | 98.44 | *98.87* | **99.49** |
| 8 | 80.00 | 97.93 | 97.71 | *98.67* | **99.75** |
| 9 | 82.53 | 98.56 | 98.76 | *98.93* | **99.92** |
| 10 | 86.28 | 97.96 | 98.07 | *98.84* | **99.83** |
| Mean | 83.26 | 98.14 | 98.07 | *98.66* | **99.77** |
| SD | 2.602 | 0.524 | 0.535 | *0.401* | **0.124** |

Figures in bold and italic indicate the best and second-best values in the row, whereas figures with an underline show the worst values.

**Table 4. Mean and standard deviation of classification accuracy of our multi-view models when trained on 80% high-quality images and tested on 20% high-quality images.**

| Run | Accuracy (%) | | | |
|---|---|---|---|---|
| | (A) | (B) | (C) | *Classifier* |
| 1 | 94.86 | *96.00* | 93.82 | **99.25** |
| 2 | 95.17 | 95.79 | *96.68* | **99.78** |
| 3 | 96.53 | *98.19* | 98.19 | **99.51** |
| 4 | 96.31 | *97.50* | 93.97 | **99.03** |
| 5 | 92.69 | 96.11 | *97.09* | **99.51** |
| 6 | 97.30 | *97.96* | 97.71 | **99.33** |
| 7 | 94.75 | 94.89 | *95.57* | **99.36** |
| 8 | 95.96 | *98.06* | 97.70 | **99.88** |
| 9 | 95.73 | 97.32 | *97.80* | **98.88** |
| 10 | 98.30 | 98.99 | *99.10* | **99.88** |
| Mean | 95.76 | *97.08* | 96.76 | **99.44** |
| SD | 1.462 | *1.242* | 1.684 | **0.324** |

Figures in bold and italic indicate the best and second-best values in the row, whereas figures with an underline show the worst values.

but based on our ten tests *without fine tuning*, the average accuracy is only 83.26%. Moreover, its standard deviation is higher than any of the SDVGG16. The outcome of the SDVGG16 model is extremely promising. With our data augmentation and spatial dropout techniques, the accuracy is increased by greater than 15% compared to that of the reference model using the same dataset. The SDVGG16 with dropout rates of 0.1 and 0.2 produces comparable results, as the mean of the one with a dropout rate of 0.1 is negligibly higher, but the one with a dropout rate of 0.2 is more consistent, as indicated by a little bit less SD. On the other hand,

**Table 5. Mean and standard deviation of the classification accuracy of our proposed models, which were trained on all high-quality images but tested on 90% low-quality images.**

| Run | Accuracy (%) | | | | |
|---|---|---|---|---|---|
| | SDVGG16 | (A) | (B) | (C) | *Classifier* |
| 1 | *80.37* | 70.40 | 77.98 | 74.11 | **85.94** |
| 2 | 79.87 | 69.78 | 76.83 | 73.83 | **86.35** |
| 3 | *80.51* | 69.83 | 76.87 | 74.34 | **85.89** |
| 4 | *80.50* | 71.97 | 78.12 | 74.90 | **86.88** |
| 5 | *80.20* | 71.37 | 77.51 | 74.91 | **85.93** |
| 6 | *80.54* | 69.32 | 76.20 | 73.34 | **85.25** |
| 7 | *80.30* | 70.25 | 77.39 | 73.53 | **85.69** |
| 8 | *80.33* | 71.26 | 77.23 | 73.97 | **86.30** |
| 9 | *80.66* | 70.27 | 76.46 | 74.54 | **86.07** |
| 10 | *80.24* | 70.69 | 78.78 | 74.81 | **87.09** |
| Mean | *80.35* | 70.51 | 77.34 | 74.23 | **86.14** |
| SD | **0.212** | 0.776 | 0.755 | 0.536 | *0.516* |

Figures in bold and italic indicate the best and second-best values in the row, whereas figures with an underline show the worst values.

**Table 6. Mean and standard deviation of the classification accuracy of our proposed models, which were retrained on 80% of high- and low-quality images and tested on remaining 20% of high- and low-quality images.**

| Run | Accuracy (%) | | | |
|---|---|---|---|---|
| | **(A)** | **(B)** | **(C)** | *Classifier* |
| 1 | 96.91 | *98.49* | 95.16 | **99.73** |
| 2 | 97.13 | *98.79* | 96.54 | **99.97** |
| 3 | 97.53 | *99.08* | 96.52 | **99.82** |
| 4 | 97.25 | *99.00* | 94.71 | **98.58** |
| 5 | 96.03 | *97.62* | 95.18 | **98.50** |
| 6 | 96.66 | *97.44* | 95.27 | **99.70** |
| 7 | 96.94 | *98.11* | 95.27 | **99.87** |
| 8 | 97.35 | *98.72* | 96.79 | **99.95** |
| 9 | 97.44 | *98.77* | 96.22 | **99.98** |
| 10 | 97.38 | *98.55* | 97.17 | **99.47** |
| Mean | 97.06 | 98.46 | 95.88 | **99.56** |
| SD | **0.430** | 0.532 | 0.811 | *0.529* |

Figures in bold and italic indicate the best and second-best values in the row, whereas figures with an underline show the worst values.

outcomes become progressively worse as its dropout rates increase from 0.3 to 0.6. Thereafter, if the dropout rate is not mentioned, it is 0.2.

**Multi-view models.** In this experiment, the train and test datasets are further divided into groups of two images for dual-view inputs in order to train and test the early-, medium-, and late-combined models (Models (A), (B), and (C)). It is noted that the dropout rate is fixed at 0.2. To train the ensemble model, the train/test dataset is re-split into groups of three images that are consumed by all three models. The three models confidence scores serve as the

**Table 7. Mean and standard deviation of the classification accuracy of our proposed models, which were trained on all high-quality images but tested on 90%low-quality images, one input was blurred intentionally.**

| Run | Accuracy (%) | | | | |
|---|---|---|---|---|---|
| | SDVGG16 | (A) | (B) | (C) | *Classifier* |
| 1 | 53.80 | 63.12 | 65.17 | *73.44* | **80.68** |
| 2 | 54.53 | 62.62 | 63.78 | *73.63* | **80.99** |
| 3 | 53.38 | 63.64 | 64.40 | *73.80* | **81.92** |
| 4 | 54.38 | 62.11 | 63.74 | *73.56* | **80.51** |
| 5 | 54.30 | 64.07 | 65.87 | *75.40* | **81.33** |
| 6 | 54.63 | 62.88 | 64.85 | *73.80* | **81.11** |
| 7 | 54.85 | 63.13 | 65.39 | *73.17* | **81.39** |
| 8 | 54.99 | 63.67 | 65.00 | *74.20* | **82.34** |
| 9 | 54.81 | 64.28 | 66.01 | *73.78* | **81.79** |
| 10 | 54.66 | 63.60 | 64.11 | *73.50* | **81.73** |
| Mean | 54.43 | 63.31 | 64.83 | *73.83* | **81.38** |
| SD | **0.474** | 0.633 | 0.771 | 0.584 | *0.547* |

Figures in bold and italic indicate the best and second-best values in the row, whereas figures with an underline show the worst values.

**Table 8. Mean and standard deviation of the classification accuracy of our proposed models, which were trained on all high-quality images but tested on 90%low-quality images, two inputs were blurred intentionally.**

| Run | Accuracy (%) | | | |
|---|---|---|---|---|
| | **(A)** | **(B)** | **(C)** | *Classifier* |
| 1 | 40.99 | 49.35 | 50.95 | **73.31** |
| 2 | 40.41 | 47.14 | 51.01 | **70.54** |
| 3 | 40.80 | 47.84 | 50.46 | **73.04** |
| 4 | 41.70 | 46.56 | 49.75 | **70.78** |
| 5 | 40.60 | 47.61 | 49.85 | **71.37** |
| 6 | 40.91 | 48.01 | 50.24 | **72.53** |
| 7 | 41.78 | 48.96 | 49.68 | **71.78** |
| 8 | 40.60 | 48.00 | 49.80 | **72.38** |
| 9 | 41.19 | 47.80 | 51.37 | **71.40** |
| 10 | 41.03 | 47.84 | 50.54 | **72.54** |
| Mean | 41.00 | 47.91 | 50.36 | **71.97** |
| SD | **0.43** | 0.76 | *0.57* | 0.89 |

Figures in bold and italic indicate the best and second-best values in the row, whereas figures with an underline show the worst values.

ensemble's inputs. After being trained, *the classifier*, which is composed of the three dual-view models and the ensemble model, predicts the species using the unseen test dataset as its inputs. In Table 3, the accuracy results of *the reference model*, Models (A), (B), (C), and *the classifier* are shown.

Table 3 demonstrates that the accuracy of the dual-view models using the same dataset proved to be significantly higher than that of *the reference model*. In particular, the results of *the classifier*, which are the outputs of the three models plus that of the ensemble, are nearly perfect and has a standard deviation of only 0.124 percent. This demonstrates that our proposed methods increased accuracy and decreased uncertainty caused by various inputs (lower SD). It should be noted that all of our proposed models are not fine-tuned, so they can be retrained without the assistance of AI specialists when new data becomes available.

## Experiments on our dataset

To show that our models do not depend on a specific dataset, all of our proposed models again attempted to guess the species using the mosquito images from the dataset produced in this study. *The reference dataset* contained five vector species and a non-vector counterpart, but our dataset contained six vector species and a non-vector counterpart.

**Accuracy tests on high-quality images.** To recheck the accuracy on another high-quality dataset, all dual-input models were initialized with ImageNet parameters, and only high-quality images from the first setting were used as input. Once more, samples were chosen at random for train/test datasets with a ratio of 8:2. In order to train and evaluate the early-, medium-, and late-combined models, the samples were further divided into groups of two images for dual-view inputs. The dropout rate was 0.2. In Table 4, the classification accuracy results for our four models are displayed.

The repeated experiments on our dataset yielded similar results. The average accuracy of the three multi-view models decreased slightly, but all were found to be over 95% with a standard deviation of less than 2%. The lower accuracy may be due to the increased number of

species (from 6 to 7) that the models must differentiate. However, the ensemble/classifier results are still nearly perfect and extremely consistent (SD = 0.324%). This demonstrates that our proposed models do not depend on a specific dataset.

**Accuracy tests on low-quality images.** In this experiment, one of the ten trained models from the previous experiment was randomly selected and retrained using all of the high-quality images. Only 20% of the input images were unseen, so the model was trained in a few epochs. To evaluate accuracy, ninety percent of the low-quality images (from the second setting) were randomly selected and used as the testing input for the models, which were obtained by retraining. In Table 5, the classification accuracy of the ten tests conducted is shown.

It is not surprising that overall precision would decline. However, the results demonstrate the robustness of the proposed models when tested on datasets that are comparable but inferior. The models can still perform, albeit, with diminished precision. The ensemble's mean accuracy was found to be 86.14 percent, which is still quite high. This experiment reveals the classifier's true potential. In previous experiments, the difference in accuracy was only a few percentage points. However, in this experiment, the ensemble's accuracy is seen to be significantly higher than that of the dual-view models. The SDVGG16 model proved to be more accurate than the dual-view models by 3–10% but less accurate than *the classifier* by 6%.

**Accuracy tests after retraining.** The aim of this experiment is to check if the proposed models can be retrained to learn more mosquito images, captured from various conditions. In this experiment, models trained with 80% of high-quality images would be retrained using ten random selections of 80% of both high- and low-quality images. All of the unselected images–both the high- and low-quality ones–would be used to evaluate the retrained models. Table 6 displays the accuracy together with means and standard deviations after the retraining.

The results presented in Tables 5 and 6 suggest that the classification accuracy after deployment may not be high immediately because the models were only trained on lab-quality images and the image quality of the photographs submitted by prospective users cannot be guaranteed. However, after tedious labeling and retraining, the accuracy of the models was restored. We hope that after a few retrainings, their inference accuracy will decrease less as they are exposed to more *generic* images.

**Robustness tests.** In this experiment, all models were trained on high-quality images before being tested on poor-quality photographs. Similar to a previous experiment, but one or two input images were purposefully blurred to diminish their quality. Thus, the one or two input images were degraded using a 2-D Gaussian filter with a kernel size of $11 \times 11$ (note that the size of images in the dataset is $512 \times 512$) and a sigma of 1.5 (the higher the sigma, the blurrier the image). In Fig 2, the difference in image quality before and after blurring is quite evident, as seen in the middle and right images. It is noted that the SDVGG16 model only accepts a single image. Dual-view models require two images, and *the classifier* (a combination of the dual-view and ensemble models; Fig 10) requires three images. Therefore, only dual-view models and *the classifier* would be tested with two bad inputs.

When the single-view SDVGG16's only input is of poor quality, the accuracy of the model drops considerably from 80.35% to 54.43% on average. In contrast, the accuracy of the multi-view models drops much less, as evidenced by a comparison of the results in Tables 5 and 7. In particular, only 0.4 percent is lost for the late-combined model (Model (C)). *The classifier* is still found to be the most reliable because two out of its three inputs are good.

The results in Table 8 demonstrate that the dual-view models perform poorly when both of their inputs are of bad quality. The classifier's accuracy still holds up at about 72% despite the fact that only one of its three inputs is good.

## Conclusion

Herein, using multi-view and spatial dropout techniques, the performance of the well- known VGG16 was enhanced. Each max pooling layer of the VGG16 was followed by a spatial drop-out layer; its architecture was modified to allow the model to simultaneously accept two images. Furthermore, an ensemble model was proposed to receive the results from the three models and yield the most accurate answer. A reference dataset containing mosquito vectors in South Korea and our own dataset containing vectors in Thailand were utilized. Using the reference dataset, the ten-run accuracy increased from 83.26% with a standard deviation of 2.602% to 99.77% with a standard deviation of 0.124%. Our dataset's classification accuracy exceeded 99% with a small standard deviation (Table 4). When low-quality images were applied, our models proved to be retrainable and robust. In this study, the classifier system is only in its prototype stage. In the real-world operation, classification accuracy will not be this high as the quality of images from users nationwide can vary hugely. Hence, producing models that are more robust to generic image quality will be our aim in further work. The ensemble model in this study combines only the results from the dual-view models, but the results from the SDVGG16 can be utilized too. Combining results from various models to increase robustness even further will also be our objective.

## Supporting information

**S1 Notebooks. Python codes and simulation results.** All Python codes and simulation results that produce the figures in Tables 2–8 are accessible from the URL given below: https://github.com/Wanchalerm-Pora/Mosquitoes-TH/tree/main/iPyNB.
(TXT)

**S1 Dataset. Dataset produced in this study.** The dataset can be accessed via the URL given below: https://www.kaggle.com/datasets/cyberthorn/chula-mosquito-classification.
(TXT)

## Author Contributions

**Conceptualization:** Wanchalerm Pora, Padet Siriyasatien, Narissara Jariyapan.

**Data curation:** Wanchalerm Pora, Natthakorn Kasamsumran, Katanyu Tharawatcharasart, Rinnara Ampol, Narissara Jariyapan.

**Formal analysis:** Padet Siriyasatien, Narissara Jariyapan.

**Funding acquisition:** Wanchalerm Pora, Narissara Jariyapan.

**Investigation:** Wanchalerm Pora, Rinnara Ampol, Narissara Jariyapan.

**Methodology:** Wanchalerm Pora, Natthakorn Kasamsumran, Katanyu Tharawatcharasart, Rinnara Ampol, Narissara Jariyapan.

**Project administration:** Narissara Jariyapan.

**Software:** Wanchalerm Pora, Natthakorn Kasamsumran, Katanyu Tharawatcharasart.

**Supervision:** Padet Siriyasatien, Narissara Jariyapan.

**Validation:** Wanchalerm Pora, Narissara Jariyapan.

**Writing – original draft:** Wanchalerm Pora, Natthakorn Kasamsumran, Katanyu Tharawatcharasart, Rinnara Ampol, Padet Siriyasatien, Narissara Jariyapan.

**Writing – review & editing:** Wanchalerm Pora, Natthakorn Kasamsumran, Katanyu Tharawatcharasart, Rinnara Ampol, Padet Siriyasatien, Narissara Jariyapan.

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
