## [Decision Letter · Decision Letter 0]

6 Feb 2023

PONE-D-22-32464Enhancement of VGG16 model with multi-view and spatial dropout for classification of mosquito vectorsPLOS ONE

Dear Dr. Pora,

Thank you for submitting your manuscript to PLOS ONE. After careful consideration, we feel that it has merit but does not fully meet PLOS ONE’s publication criteria as it currently stands. Therefore, we invite you to submit a revised version of the manuscript that addresses the points raised during the review process.

We look forward to receiving your revised manuscript.

Kind regards,

Anwar P.P. Abdul Majeed

Academic Editor

PLOS ONE

Journal Requirements:

"This study is funded by Thailand Science Research and Innovation Fund Chulalongkorn University (CU FRB65 hea(45) 052 30 33)."

"This study is funded by Thailand Science, Research and Innovation Fund Chulalongkorn University (CU_FRB65_hea(45)_052_30_33). This fund is managed by Thailand science, research and innovation office (https://www.tsri.or.th).  Its principal investigator and investigator are Narissara Jariyapan and Wanchalerm Pora, respectively.   

5. We note that you have referenced (ie. Jomtarak R, Kittichai V et al. [17]) which has currently not yet been accepted for publication. Please remove this from your References and amend this to state in the body of your manuscript: (ie "Jomtarak R, Kittichai V  et al. [Unpublished]") as detailed online in our guide for authors:

Reviewers' comments:

Reviewer's Responses to Questions

**Comments to the Author**

1. Is the manuscript technically sound, and do the data support the conclusions?

Reviewer #1: Yes

2. Has the statistical analysis been performed appropriately and rigorously? 

Reviewer #1: Yes

3. Have the authors made all data underlying the findings in their manuscript fully available?

Reviewer #1: Yes

4. Is the manuscript presented in an intelligible fashion and written in standard English?

Reviewer #1: Yes

5. Review Comments to the Author

Reviewer #1: Thank you for the interesting study. The manuscript has been presented well. However, there are improvements that can be made.

Would you please mention the resolution of the high-quality photographs mentioned in the subchapter “Image datasets”?

What is the resolution of the photographs captured by the mid-range and low-end smartphones mentioned in the subchapter “Image datasets”?

This may or may not affect the input image details as it will be resized accordingly, but it is worth to be mentioned.

Thank you.

6. PLOS authors have the option to publish the peer review history of their article (what does this mean?). If published, this will include your full peer review and any attached files.

Reviewer #1: No

---

## [Author Response · Author response to Decision Letter 0]

28 Feb 2023

There is only one reviewer and (s)he gave only two comments:

Comment 1. Would you please mention the resolution of the high-quality photographs mentioned in the subchapter "Image datasets"?

Response: Thank you so much for bringing this to our attention. We added a sentence: The 64MP smartphone was set to produce photos at its highest resolution of 9,248 x 6,936. Hopefully, it will provide additional information to the audience. 

Comment 2. What is the resolution of the photographs captured by the mid-range and low-end smartphones mentioned in the subchapter "Image datasets"?

Response: Again, thank you for pointing this out. So we added a sentence: "The 64MP mid-range and the 13MP low-end phones produced images with resolutions of 9,248 x 6,936 and 4,128 x 3,096, respectively." To remind the audience that the quality of the image does not depend solely on the image resolution, we added further a sentence: "Note that the quality of the lens and image sensors on the phones' cameras has a greater impact on the resulting photos than the resolutions of the cameras." Despite having the same image resolution, the Vivo V21 (the high-end) and Samsung A52s (the mid-range) received DXOMark (still camera) scores of 105 and 88, respectively. This shows that the cameras on the Vivo V21 are generally better than those on the Samsung A52s. Note that DXOMark.com did not give the Vivo Y21 (the low-end) a camera test.

---

## [Decision Letter · Decision Letter 1]

28 Mar 2023

Enhancement of VGG16 model with multi-view and spatial dropout for classification of mosquito vectors

PONE-D-22-32464R1

Dear Dr. Pora,

We’re pleased to inform you that your manuscript has been judged scientifically suitable for publication and will be formally accepted for publication once it meets all outstanding technical requirements.

Kind regards,

Anwar P.P. Abdul Majeed

Academic Editor

PLOS ONE

Additional Editor Comments (optional):

Reviewers' comments:

Reviewer's Responses to Questions

**Comments to the Author**

1. If the authors have adequately addressed your comments raised in a previous round of review and you feel that this manuscript is now acceptable for publication, you may indicate that here to bypass the “Comments to the Author” section, enter your conflict of interest statement in the “Confidential to Editor” section, and submit your "Accept" recommendation.

Reviewer #1: All comments have been addressed

2. Is the manuscript technically sound, and do the data support the conclusions?

Reviewer #1: Yes

3. Has the statistical analysis been performed appropriately and rigorously? 

Reviewer #1: Yes

4. Have the authors made all data underlying the findings in their manuscript fully available?

Reviewer #1: Yes

5. Is the manuscript presented in an intelligible fashion and written in standard English?

Reviewer #1: Yes

6. Review Comments to the Author

Reviewer #1: Thank you for the responses. All comments has been addressed. Please prepare the manuscript for the final proof by the publisher.

7. PLOS authors have the option to publish the peer review history of their article (what does this mean?). If published, this will include your full peer review and any attached files.

Reviewer #1: No

---

## [Editor Report · Acceptance letter]

4 Apr 2023

PONE-D-22-32464R1 

Enhancement of VGG16 model with multi-view and spatial dropout for classification of mosquito vectors  

Dear Dr. Pora:

I'm pleased to inform you that your manuscript has been deemed suitable for publication in PLOS ONE. Congratulations! Your manuscript is now with our production department. 

Kind regards, 

on behalf of

Dr. Anwar P.P. Abdul Majeed 

Academic Editor

PLOS ONE